# Efficient Anatomy Segmentation in Laparoscopic Surgery using Multi-Teacher Knowledge Distillation

**Lennart Maack** [1]                  LENNART.MAACK@TUHH.DE
**Finn Behrendt** [1]                  FINN.BEHRENDT@TUHH.DE
**Debayan Bhattacharya** [1]            DEBAYAN.BHATTACHARYA@TUHH.DE
**Sarah Latus** [1]                    S.LATUS@TUHH.DE
**Alexander Schlaefer** [1]              SCHLAEFER@TUHH.DE

[1] *Institute of Medical Technology and Intelligent Systems, Hamburg University of Technology, Hamburg, Germany*

**Editors:** Accepted for publication at MIDL 2024

## Abstract

Automatic segmentation of anatomical structures in laparoscopic images or videos is an important prerequisite for visual assistance tools which are designed to increase efficiency and safety during an intervention. In order to be used in a realistic clinical scenario, both high accuracy and real-time capability are required. Current deep learning networks for anatomy segmentation show high accuracy, but are not suitable for real-time clinical application due to their large size. As smaller, real-time capable deep learning networks show lower segmentation performance, we propose a multi-teacher knowledge distillation approach applicable to partially labeled datasets. We leverage the knowledge of multiple anatomy-specific, high-accuracy teacher networks to improve the segmentation performance of a single and efficient student network capable of segmenting multiple anatomies simultaneously. To do so, we minimize the Kullback-Leibler divergence between the normalized anatomy-specific teacher logits and the respective normalized logits of the student. We conduct experiments on the Dresden Surgical Anatomy Dataset, which provides multiple subsets of binary segmented anatomical structures. Results show that our approach can increase the overall Dice score for different real-time capable network architectures for anatomy segmentation.

**Keywords:** Anatomy Segmentation, Real-Time, Surgical Computer Vision, Knowledge Distillation

## 1. Introduction

Postoperative complications remain a major problem for both the healthcare system and the individual patient and are associated with higher healthcare costs and poorer outcomes (Dencker et al., 2021). An important factor in decreasing postoperative complications is the reduction of technical errors, which are defined as adverse events directly related to manual errors of the surgeon (Fecso et al., 2017; Suliburk et al., 2019).

The increasing adoption of minimally invasive procedures, which rely on the visualization by endoscopic cameras, leads to an increasing amount of available surgical video data. This enables the data-driven analysis of surgical video data using computer vision techniques, such as deep learning, to visually assist surgeons (Mascagni et al., 2022). An essential prerequisite for computer vision-based surgical assistance applications is the accurate and real-time

perception of the intraoperative environment, e.g. by segmenting the surgical instruments and anatomical structures. To evaluate the accuracy of computer vision-based methods for anatomy segmentation, various data sets with pixel-wise annotations of anatomical structures were introduced (Bamba et al., 2021; Madani et al., 2020; Allan et al., 2020). As these data sets have little diversity due to their size or are based solely on simpler porcine tissue data, the applicability of deep learning models trained with such data sets in a real clinical setting is limited. Therefore, Carstens et al. (Carstens et al., 2023) published the largest public data set of laparoscopic images to date, namely the Dresden Surgical Anatomy Dataset (DSAD). It is divided into partially annotated sub-datasets, containing overall $13,195$ laparoscopic images with pixel-wise annotations of eleven anatomical structures. As only binary annotations for one anatomy are provided in a sub-dataset, although several other anatomical structures without annotation are visible, complete information about the background remains unknown. To tackle the problem of training networks with partially labeled datasets, the usage of annotation adaptive loss functions has been proposed (Vu et al., 2021; Ulrich et al., 2023). In their recent work, Kolbinger et al. (Kolbinger et al., 2023) trained a combined network with a shared encoder and multiple decoders for each of the sub-datasets in the DSAD. In addition, they made use of mutual-exclusivity by incorporating the information of a positive annotation of one class as a negative annotation for all other classes. Despite achieving segmentation accuracy comparable to human experts, there are two limitations. First, the binary segmentation accuracy of the combined network is inferior compared to anatomy-specific single-encoder, single-decoder networks (Kolbinger et al., 2023). Second, due to the large size of the encoder and decoder networks, achieving a sufficiently high frame rate for delay-free segmentation of anatomical structures on dedicated hardware in the operating room can not be guaranteed. Methods for real-time capable segmentation in surgical videos include the development of lightweight convolution-based architectures for faster segmentation of surgical instruments in laparoscopy (Tomasini et al., 2022; Pakhomov and Navab, 2020; Jha et al., 2021). While networks with a low amount of parameters can provide good segmentation performance for simpler tasks such as surgical instrument segmentation, they are ineffective at learning complex features required for accurate segmentation of anatomies in surgical videos. Knowledge Distillation has drawn attention to overcome the dilemma of decreasing performance when using smaller models capable of faster inference speed. Xie et al. (Xie et al., 2018) transfer the zero- and first-order knowledge from a strong teacher network to guide the fast student network. Qin et al. (Qin et al., 2021) improve the segmentation performance on public CT datasets by proposing a novel module that encodes regional knowledge for a student network. To overcome the dependency of student networks on a single teacher network, Amirkhani et al. (Amirkhani et al., 2021) include multiple teacher networks trained with the same input but different style transfers and data augmentations.

In this work, we propose a multi-teacher knowledge distillation (MT-KD) approach that leverages the knowledge of multiple anatomy-specific, high-accuracy teacher networks to tackle the problem of training a single network with partially labeled datasets. Specifically, we use MT-KD to improve the segmentation performance of a real-time capable student network with a small number of parameters. In a first step, multiple teacher networks are trained to obtain high anatomy-specific accuracy. In a second step, the Kullback-Leibler (KL) divergence is minimized between the normalized output logits of the individ-

ual anatomy-specific teacher models and the normalized output logits of the corresponding anatomy-specific decoder of the student model. This way, the teacher networks guide the student to pay more attention to the most salient regions in order to accurately segment the anatomies. By improving the segmentation accuracy of small student networks, capable of segmenting multiple anatomies simultaneously, we aim to increase the applicability of computer vision methods in realistic surgical scenarios. The comprehensive evaluation of our MT-KD approach shows increased segmentation performance across various network architectures with a small number of parameters, as shown in Figure 1.

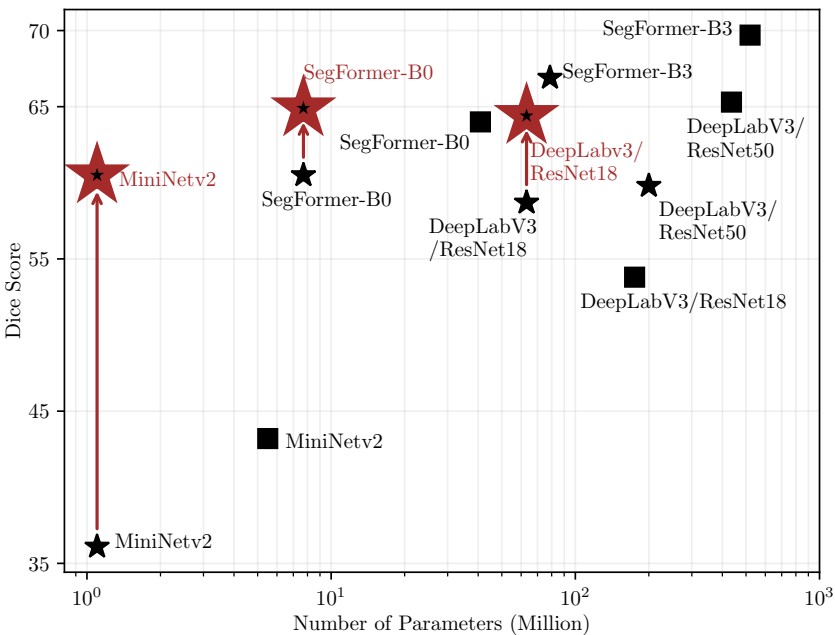

Figure 1: Performance of different segmentation networks presented as the mean Dice score over all eleven anatomies in the DSAD. The red stars indicate the performance of the network trained with Multi-Teacher Knowledge Distillation (MT-KD). The red arrows indicate the respective improvement. ★ refers to models using a common encoder and eleven anatomy-specific decoders. ■ indicates segmentation models that are trained on each anatomy separately.

## 2. Method

Our multi-teacher knowledge distillation approach with its two stages is schematically illustrated in Figure 2 and explained in more detail in the following.

**Stage 1:**
During the first stage, $D = \{A_{i=1}, A_{i=2}, .., A_{i=N}\}$ corresponds to the overall data set, where

$A_i$ denotes the anatomy-specific data set of the $i^{th}$ anatomy of $N$ anatomies. In the following, only one of the anatomy-specific data sets is considered.

Let $A$ be denoted as $A = \{(x_j, y_j)\}_{j=1}^{M}$, where $x_j \in \mathbb{R}^{3 \times H \times W}$ and $y_j \in \mathbb{R}^{1 \times H \times W}$ correspond to the input image and binary segmentation mask, respectively. $M$ denotes the overall number of images in $A$. Further, a teacher segmentation model $T$, consisting of one encoder $F^{enc}$ and one decoder $F^{dec}$, is trained using the standard pixel-wise binary cross entropy loss function formulated as:

$$\mathcal{L}_{CE} = -\sum_{j=1}^{M} \sum_{k=1}^{H \times W} [y_{j,k} \log(T(x_{j,k})) + (1 - y_{j,k}) \log(1 - T(x_{j,k}))] \tag{1}$$

**Stage 2:**

In this stage, the same data set $D$ is utilized as in stage 1. This time, a student segmentation model $S$ that consists of one encoder $F^{enc}$ and $N$ anatomy-specific decoders $\{F_1^{dec}, F_2^{dec}, .., F_N^{dec}\}$ is optimized using two different objective functions.

For the first objective function, we follow the description from Kolbinger et al. (Kolbinger et al., 2023). There, the pixel-wise binary cross entropy loss according to Equation 1 is calculated separately using the output probabilities of each of the $N$ anatomy-specific decoders. In detail, the loss is calculated for each pixel, only if the annotated anatomy $i$ in the input image corresponds to the respective anatomy-specific decoder $F_i^{dec}$ of $S$. For all other decoders $F_{\neq i}^{dec}$, only the pixels in $x_j$ that belong to the anatomy $i$ are considered for the loss calculation as the false positive class, as shown in Figure 2. The remaining pixels are not considered for the loss calculation, as only binary segmentation masks are used in DSAD and several anatomies can appear per image and it cannot be ruled out that all other pixels do not contain the anatomy $i$.

The second objective function utilizes the anatomy-specific knowledge of the various teacher models with frozen parameters from stage 1. Similar as in work by Shu et al. (Shu et al., 2021), where the normalized activations of corresponding channels between the teacher and student network are aligned using the KL divergence, we utilize the normalized output logits of each of the anatomy-specific teacher models and minimize the discrepancy to the normalized output logits of the corresponding anatomy-specific decoder of the student model. As the logits of a well-trained, anatomy-specific teacher model generally show salient anatomical regions, the student model, capable of segmenting multiple anatomies simultaneously, can be guided. This results in overall higher segmentation performance of the student. Let $z_{i,C}^T$ and $z_{i,C}^S$ be the output logits of the anatomy-specific teacher model $T_i$ and the student decoder $F_i^{dec}$ of anatomy $i$, with $C$ being either the anatomy or false positive/background class. First, the output logits $z_{i,C}^T$ and $z_{i,C}^S$ are divided by a temperature value $\mathcal{T}$ and then normalized using the softmax function $\sigma(z) = \frac{e^z}{\sum(e^z)}$. The temperature value $\mathcal{T}$ is used to control the softness of the probability distribution. Second, to evaluate the discrepancy between the two probability distributions $p_{i,C}^T = \sigma(\frac{z_{i,C}^T}{\mathcal{T}})$ and $p_{i,C}^S = \sigma(\frac{z_{i,C}^S}{\mathcal{T}})$, we utilize the KL divergence (Kullback and Leibler, 1951).

The second objective function during stage 2 can therefore be denoted as:

$$\mathcal{L}_{KL_i}(p_{i,C}^S, p_{i,C}^T) = p_{i,C}^T \cdot \log(\frac{p_{i,C}^T}{p_{i,C}^S}) \tag{2}$$

With $\lambda$ as a weighting parameter, the overall objective function of stage 2 can be formulated as:

$$\mathcal{L} = \sum_{i=1}^{N} \mathcal{L}_i = \sum_{i=1}^{N} \mathcal{L}_{CE_i} \cdot \lambda \mathcal{L}_{KL_i} \tag{3}$$

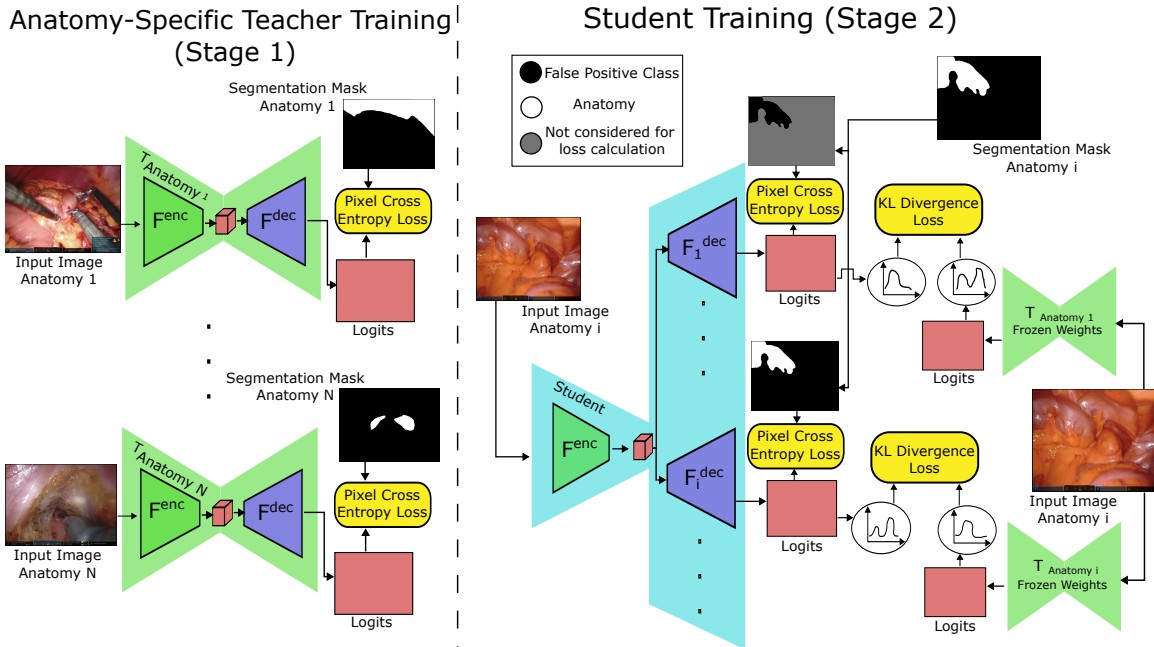

Figure 2: Schematic of the proposed MT-KD approach. In stage 1, multiple teacher networks are trained to obtain high anatomy-specific accuracy. In stage 2, the normalized output logits of the individual anatomy-specific teacher models are used to guide the training of the student models using the KL-divergence.

## 3. Experimental Setup

### 3.1. Data

To evaluate our MT-KD approach, we use the DSAD that consists of $13,195$ high-quality laparoscopic images with pixel-wise annotations of eleven intra-abdominal anatomical structures, i.e., abdominal wall, colon, inferior mesenteric artery, intestinal veins, liver, pancreas, small intestine, spleen, stomach, ureter and vesicular glands (Carstens et al., 2023). We follow the training, validation and test splits as well as the pre-processing steps and augmentations as in the original work from (Kolbinger et al., 2023).

### 3.2. Implementation details

For all experiments with the proposed approach, we utilize the Adam optimizer (Kingma and Ba, 2014), using a learning rate of 5e-4 during stage 1 and a learning rate of 1e-3 for

stage 2. Additionally, an exponential learning rate scheduler is used. We train our models for 100 epochs and 60 epochs in stage 1 and 2 and end up with a final learning rate of 1.5e-6 and 2.5e-5, respectively. During the teacher-student knowledge distillation, we follow the implementation details from (Shu et al., 2021) and use the temperature value $\mathcal{T} = 4.0$ for the calculation of the KL divergence and the weighting parameter $\lambda = 3$. During all training experiments, a batch size of 8 and an input image size of $640 \times 512$ is utilized. The utilized convolutional encoder architectures are pretrained on either the COCO or Cityscapes data set (Lin et al., 2014; Cordts et al., 2016). For the transformer-based segmentation networks, i.e., SegFormer, we use pretrained weights from the Cityscape data set. For the evaluation on the test data set, the model that obtained the best results in terms of the Dice score on the validation set is used. We implement all models in Pytorch [1].

## 4. Results

To evaluate the proposed approach in this work, the mean Dice score among all eleven anatomies in the DSAD is used. Furthermore, the number of parameters of each segmentation model as well as the inference time in form of time for computing the segmentation mask of one input image of size $1280 \times 720$ (High Definition) is determined.

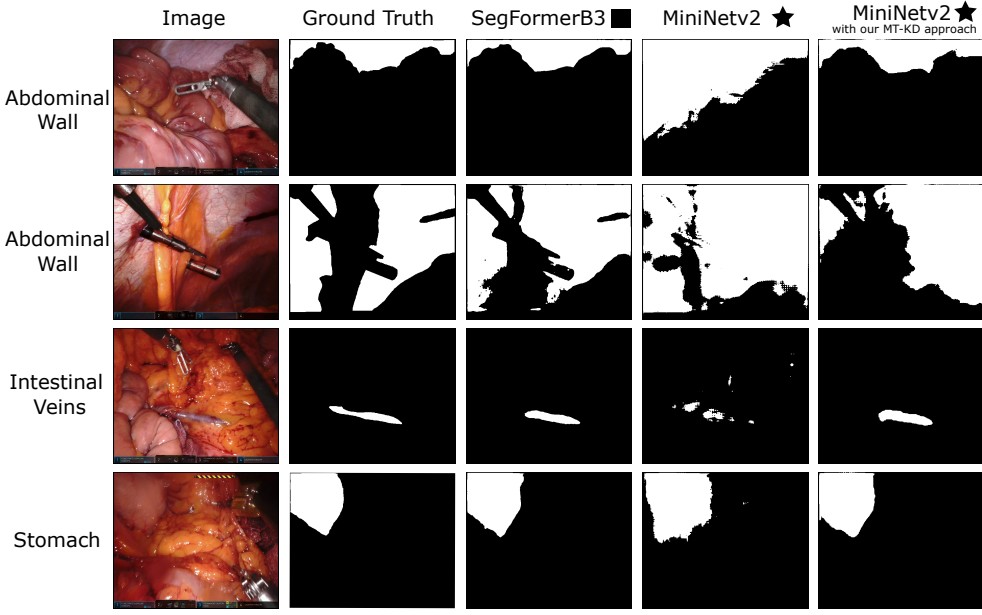

Figure 3: Qualitative segmentation results for different anatomies.

In order to assess the improvements of our the approach, we evaluate previous deep learning models applied to the DSAD regarding segmentation accuracy and inference time, as well as another state-of-the-art network for efficient surgical segmentation, as shown in

---

1. Code available at:
   https://github.com/lennart-maack/Efficient-Anatomy-Segmentation-w-Multi-Teacher-KD

Table 1: Segmentation and inference speed results. ■ indicates segmentation models that are trained on each anatomy separately, consisting of one encoder and one decoder. ★ indicates models using a common encoder and eleven anatomy-specific decoders. Dice score (%) is calculated as the mean over all eleven anatomies. Time corresponds to the inference time in ms for one image ($1280 \times 720$) on a NVIDIA RTX3090. SegFormer-B3 ■ is used as a teacher network for our Multi-Teacher Knowledge Distillation (MT-KD) approach.

| Architecture | Encoder | Params(M) | Dice ($\uparrow$) | Time ($\downarrow$) | FPS ($\uparrow$) |
|---|---|---|---|---|---|
| DeepLabv3 ■ | ResNet18 | 174.8 | 53.8 | 160 | 6 |
| (Chen et al., 2017) | ResNet50 | 435.9 | 65.3 | 367 | 3 |
| | EfficientNetb0 | 80.3 | 62.8 | 262 | 4 |
| | EfficientNetb3 | 159.1 | 66.3 | 479 | 2 |
| SegFormer ■ | SegFormerB0 | 40.8 | 64.0 | 180 | 6 |
| (Xie et al., 2021) | SegFormerB3 | 519.5 | **69.7** | 548 | 2 |
| MiniNetv2 ■ (Tomasini et al., 2022) | MiniNetv2 | 5.5 | 43.2 | 35 | 28 |
| | ResNet18 | 63.1 | 58.7 | 42 | 23 |
| DeepLabv3 ★ | ResNet50 | 200.0 | 59.8 | 112 | 9 |
| (Chen et al., 2017) | EfficientNetb0 | 40.0 | 60.3 | 45 | 22 |
| | EfficientNetb3 | 52.2 | 64.2 | 67 | 15 |
| SegFormer ★ | SegFormerB0 | 7.7 | 60.5 | 39 | 25 |
| (Xie et al., 2021) | SegFormerB3 | 78.7 | 66.9 | 153 | 6 |
| MiniNetv2 ★ (Tomasini et al., 2022) | MiniNetv2 | 1.1 | 36.1 | **26** | **38** |
| DeepLabv3 ★ | EfficientNetb0 | 40.0 | 64.5 | 45 | 22 |
| (with our MT-KD approach) | ResNet18 | 63.1 | 64.4 | 42 | 23 |
| SegFormer ★ (with our MT-KD approach) | SegFormerB0 | 7.7 | 64.9 | 39 | 25 |
| MiniNetv2 ★ (with our MT-KD approach) | MiniNetv2 | 1.1 | 60.5 | **26** | **38** |

Table 1. Both anatomy-specific models with one encoder and one decoder (■), as well as models with one common encoder and multiple anatomy-specific decoders (★) are evaluated. From the results in Table 1, we observe that segmentation models trained on each anatomy separately outperform networks with the same architecture but using a common encoder and eleven anatomy-specific decoders in terms of Dice score. However, the number of parameters increases significantly when using multiple anatomy-specific models which leads to low inference speed. Furthermore, the results show superior performance in terms of Dice score when DeepLabv3 is used with EfficientNet as an encoder compared to ResNet encoders. Small network architectures such as MiniNetv2 enable real-time capabilities, but show significantly lower segmentation performance. The transformer-based architectures SegFormer achieves higher segmentation accuracy in comparison to convolutional-based

DeepLabv3 networks. Our proposed MT-KD approach increases the segmentation performance of both convolutional-based segmentation networks and transformer-based segmentation networks. The most significant increase due to our MT-KD approach can be shown for small models, i.e. MiniNetv2. In this case, the Dice score increases from 36.1% to 60.5%. Qualitative results in Figure 3 show more accurate segmentation of anatomies for MiniNetv2 when trained with our MT-KD. Especially for smaller details, the segmentation accuracy can be increased by using large and accurate teacher models. A uniform increase in segmentation performance can be observed across all eleven anatomies. The specific segmentation results in terms of Dice score for individual anatomies can be found in the Appendix.

## 5. Discussion and Conclusion

Current deep learning networks for anatomy segmentation suffer from two problems. Either they show good segmentation performance, similar to human experts, but are too large for real-time applications or they are efficient enough for real-time applications but do not show sufficient performance for high-accuracy segmentation. In this work, we propose a multi-teacher knowledge distillation (MT-KD) approach that leverages the knowledge of multiple anatomy-specific, high-accuracy teacher networks to tackle the problem of training a single and efficient network with partially labeled datasets. By minimizing the discrepancy between the normalized logits of anatomy-specific, high-accuracy teacher networks and a single and efficient student network, the segmentation accuracy of various small, real-time capable network architectures is improved while retaining high inference speed.

Our results demonstrate highest segmentation performance with 66.3% and 69.7% Dice score for anatomy-specific, high capacity teacher networks such as DeepLabv3/Eff.Netb3 ■ and SegFormerB3 ■. We assume higher capacity to learn complex, anatomy-specific features. The results further demonstrate that anatomy-specific, low capacity networks such as MiniNetv3 ■ only achieve an overall Dice Score of 43.2%, failing to learn valuable features in the data. For application in realistic surgical scenarios, anatomy-specific models need to be operated sequentially in order to segment several anatomies. A combined architecture with one common encoder and anatomy-specifc decoders enables simultaneous anatomy segmentation. Although the segmentation performance of the combined architecture decreases only by 2.1% and 2.8%, for DeepLabv3/EfficientNetb3 ★ and SegFormerB3 ★, respectively, models with large encoder and decoder architectures still do not achieve a sufficiently high frame rate (6 FPS and 15 FPS). With our approach, the segmentation performance of models with smaller and thus faster architectures increases by up to 24.4%. Especially, the accurate segmentation of smaller and more complex anatomies can be improved for smaller segmentation networks when guided by accurate teacher models with our MT-KD approach. Overall, the segmentation accuracy of high-capacity, anatomy-specific networks remains at least 1.8% higher, however, almost two orders of magnitude fewer parameters are required. We evaluated our approach on the recently published DSAD. In order to evaluate whether the generalization ability of smaller student models changes similarly to that of larger teacher models when applied to other laparoscopic data sets, it is of interest to perform further comprehensive studies on different laparoscopic data sets.

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

# Appendix A. Anatomy specific results

| Architecture | Encoder | Abdominal wall | Colon | Inferior mesenteric artery | Intestinal veins | Liver | Pancreas | Small intestine | Spleen | Stomach | Ureter | Vesicular glands |
|---|---|---|---|---|---|---|---|---|---|---|---|---|
| DeepLabv3 ■ (Chen et al., 2017) | ResNet18 | 83 | 69 | 44 | 39 | 65 | 28 | 79 | 56 | 59 | 33 | 37 |
| | ResNet50 | 90 | 79 | 54 | 54 | 80 | 37 | 87 | 79 | 71 | 47 | 40 |
| | Eff.Netb0 | 88 | 77 | 51 | 49 | 71 | 42 | 86 | 74 | 66 | 40 | 47 |
| | Eff.Netb3 | 90 | 79 | 54 | 56 | 76 | 43 | 88 | 78 | 66 | 48 | 52 |
| SegFormer ■ (Xie et al., 2021) | SegFormerB0 | 89 | 76 | 51 | 51 | 78 | 45 | 85 | 73 | 64 | 44 | 48 |
| | SegFormerB3 | 91 | 79 | 58 | 58 | 83 | 46 | 89 | 81 | 75 | 52 | 55 |
| MiniNetv2 ■ (Tomasini et al., 2022) | MiniNetv2 | 80 | 55 | 34 | 24 | 61 | 25 | 70 | 38 | 44 | 20 | 25 |
| DeepLabv3 ★ (Chen et al., 2017) | ResNet18 | 83 | 70 | 44 | 47 | 76 | 37 | 77 | 73 | 61 | 32 | 46 |
| | ResNet50 | 83 | 72 | 49 | 47 | 68 | 35 | 79 | 73 | 63 | 35 | 47 |
| | Eff.Netb0 | 83 | 72 | 48 | 47 | 71 | 37 | 80 | 75 | 64 | 39 | 48 |
| | Eff.Netb3 | 84 | 75 | 53 | 57 | 72 | 43 | 81 | 76 | 70 | 47 | 51 |
| SegFormer ★ (Xie et al., 2021) | SegFormerB0 | 83 | 73 | 42 | 53 | 75 | 42 | 79 | 69 | 62 | 42 | 46 |
| | SegFormerB3 | 89 | 77 | 55 | 53 | 78 | 45 | 84 | 81 | 71 | 49 | 54 |
| MiniNetv2 ★ (Tomasini et al., 2022) | MiniNetv2 | 61 | 46 | 18 | 36 | 52 | 29 | 51 | 36 | 34 | 13 | 21 |
| DeepLabv3 ★ (w/ our KD-approach) | Eff.Netb0 | 86 | 74 | 55 | 54 | 69 | 41 | 84 | 81 | 68 | 48 | 49 |
| | ResNet18 | 87 | 76 | 55 | 54 | 77 | 37 | 81 | 78 | 70 | 43 | 51 |
| SegFormer ★ (w/ our KD-approach) | SegFormerB0 | 84 | 75 | 51 | 53 | 76 | 45 | 81 | 81 | 74 | 45 | 49 |
| MiniNetv2 ★ (w/ our KD-approach) | MiniNetv2 | 83 | 74 | 46 | 49 | 72 | 40 | 80 | 74 | 64 | 41 | 42 |

Table 2: Segmentation results for all individual anatomies in terms of Dice score (%).

