# OpenReview forum: "Efficient Anatomy Segmentation in Laparoscopic Surgery using Multi-Teacher Knowledge Distillation"
_MIDL.io/2024/Conference — MIDL 2024 Poster_

### Official Review · Reviewer_dvkg · 2024-02-25

**Confidence:** 4
**Preliminary Rating:** 2

**Summary:**

The study aims to enhance a specific dataset released by Carstens et al. using distillation learning. This dataset has a significant problem as all images contain one category while potentially including other categories without labels. The proposed method effectively addresses this issue by integrating information into the feature space through distillation learning, allowing the model to extract features from all categories within a single model. Although this approach is successful for datasets with similar challenges, its application is limited to such datasets, and the distillation learning method it employs may lack novelty in the field.

**Strengths:**

This paper introduces an approach to integrating features from all categories within a dataset, where images are labeled with only one type of information while potentially containing other categories without explicit labels. Through experiments and analysis, it is evident that this method enhances performance on such datasets and points towards a promising direction for handling similar data challenges in the future.

**Weaknesses:**

1. The paper primarily introduces a method utilizing distillation learning, which is considered a common approach in the field. Moreover, it lacks comparisons with other distillation learning methods and is highly specific to a particular dataset. Therefore it might have a limited contribution.
2. The experimental section would benefit from the inclusion of visualization analysis to gain a deeper understanding of how the method performs in various scenarios, particularly those that are more complex and challenging.
3. The authors conducted experiments on two recent methods and only one dataset to evaluate the proposed method, so it might lacked persuasiveness.
4. In the figures, Fig. 1 could be improved by adding color to the legends, and the readability of Fig. 2 could be improved by adding more explanation in the title.

**Detailed Comments:**

See the weakness for more details.

**Justification Of The Preliminary Rating:**

This paper addresses the specific problem in the public dataset, but it might not be general in many medical datasets. Although it does have an improvement in efficiency and effectiveness with the proposed distill learning, it has limited contribution to medical analysis considering the novelty of the method and its benefit to the community.

**Questions To Address In The Rebuttal:**

1. Highlight the novelty and contribution.
2. Improve the readability of the method, figure, and the relative description.
3. Analyse the experiment more in focus and present visualization analysis.
4. Consider the comparison methods and justify the design of the article.

---

> ### Author Response · Authors · 2024-03-18
>
> "1. Highlight the novelty and contribution"
>
> - We thank the reviewer for the important request. Our main contribution consists of using multiple teachers to increase segmentation performance of real-time capable student networks. To the best of our knowledge, we firstly propose this approach for partially labeled datasets. We conduct a comprehensive evaluation of our approach for multiple network architecturs and and show the advantages of our approach. We revised the introduction section to highlight the novelty and contribution more in depth.
>
> "2. Improve the readability of the method, figure, and the relative description."
>
> - We updated the Figures as well as relative description to improve readability.
>
> "3. Analyse the experiment more in focus and present visualization analysis."
> - We conducted additional experiments with DeepLabv3/ResNet18 and MiniNetv2. Additionally, we added anatomy-specific results in the appendix. We added qualitative results in the appendix and aim to add more qualitative results in the future.
>
> "4. Consider the comparison methods and justify the design of the article."
> - We considered comparison methods in the Introduction and aim to add associated experiments in the future.

---

### Official Review · Reviewer_uZF8 · 2024-03-03

**Confidence:** 4
**Preliminary Rating:** 3
**Recommendation:** Poster
**Final Rating:** 3.5

**Summary:**

The authors propose a teacher-student learning approach on the DSAD dataset, which enables learning from a data with imbalanced annotations. Their method allows for real-time applications with better performance than models with similar inference times.

**Strengths:**

The presented learning approach strikes a balance between model complexity (increasing inference time) and segmentation performance. They compare various baseline segmentation approaches based on inference time and average segmentation performance.

**Weaknesses:**

The significance of the improvements is difficult to assess without tests of significance. The introduction lacks important aspects of the field that should be adressed to fully understand the problem the authors wish to tackle.

**Detailed Comments:**

- Looking forward to the public code repository!
- In the introduction the authors mention that the DSAD dataset only has one segmentation per image, whereas the proposed model trains to segment all anatomies in all of the images. It would clearly show the usefulness of the trained model to see an example image where the ground truth contains only a single segmentation, however the model segments all other anatomies that are present as well. Perhaps in the appendix.

**Justification Of Final Rating:**

I would like to thank the authors for thoroughly addressing all my questions and the concerns of the other reviewers. Acknowledging other possible design choices, such as a shared decoder with multiple outputs and loss functions tackling the same issue is certainly helpful, however it would be more appropriate (but also time consuming) to compare against these alternate methods. The clarifications and the more comprehensive evaluations are much appreciated, so I am slightly increasing my rating.

**Justification Of The Preliminary Rating:**

The description of the problem and the evaluation of the proposed method could be improved to clearly show a justification of the approach and its advantages. However the learning approach has merit, and it has significance in the field of real-time anatomy segmentation.

**Questions To Address In The Rebuttal:**

- The authors describe two approaches for multi-anatomy segmentation, the first being that every anatomy has a separate encoder and decoder, while the other is that all anatomies have the same encoder and different decoders. The authors fail to address the arguably most common segmentation approach where all anatomies have the same encoder and decoder, and they are returned in a multi-channel feature map (eg. in the example of the UNet). Can the authors explain why this was not mentioned in the manuscript?
- The authors claim that the DSAD dataset cannot be used to train a combined segmentation model, since its segmentations are imbalanced (some images contain anatomies that are not segmented). However these can be addressed with loss functions customized for imbalanced datasets, eg. the data-adaptive loss function proposed in Vu et al. 2022 "A Data-Adaptive Loss Function for Incomplete Data and Incremental Learning in Semantic Image Segmentation". Can the authors discuss why the teacher models are required to train the light-weight model to handle all anatomies?
- The authors claim that while the teacher models were trained for 100 epochs, the student models required only 60. I assume these numbers were found empirically? Please explain how so. If the performance did not improve over 100 and 60 epochs respectively, can the authors address why the student models are faster to train than the teacher models?
- In the results section the authors say that "A uniform increase in segmentation performance can be observed across all eleven anatomies". However these detailed results are not presented in the manuscript. Perhaps the authors could include a more detailed evaluation table in the appendix to show anatomy-based evaluations?
- The performance of the proposed model is questionable, based on the results in Table 1. Looking at the values, a reader can say, that models that have small number of trainable parameters (MiniNetv2) perform poorly, however they have a short inference time. Whereas models that have a large number of parameters (SegFormerB3) perform well, however they have a slow inference time. What the authors propose is a model that follows this trend, it has more parameters than the MiniNet, and it performs better, and is slower, however it has fewer parameters than SegFormer, therefore it performs worse and is faster. It is difficult to see what the model adds to the list of solutions in the field. Models trained with the KD-approach doesn't stand out in any of the metrics individually, only when looking at the metrics collectively. What do the authors think of collecting the results instead only for a certain model architecture, showing how the KD-approach and the other two learning methods impact performance and inference time? What is the benefit of a collective overall table such as Table 1 for all model performances?

---

> ### Author Response · Authors · 2024-03-18
>
> "The authors describe two approaches for multi-anatomy segmentation, the first being that every anatomy has a separate encoder and decoder, while the other is that all anatomies have the same encoder and different decoders. The authors fail to address the arguably most common segmentation approach where all anatomies have the same encoder and decoder, and they are returned in a multi-channel feature map (eg. in the example of the UNet). Can the authors explain why this was not mentioned in the manuscript?"
>
> - We agree with the reviewer that the arguably most common segmentation approach is a single encoder, single decoder architecture with multi-channel output. However, we claim that because of the partially labeled sub-datasets and the missing information about the background, a single decoder would show inferior performance compared to a multi-decoder architecture. This must be empirically proven in future work. Previous work on annotation adaptive loss functions as one way to tackle the problem of partially labeled datasets has been added to the Introduction Section.
>
> "The authors claim that the DSAD dataset cannot be used to train a combined segmentation model, since its segmentations are imbalanced (some images contain anatomies that are not segmented). However these can be addressed with loss functions customized for imbalanced datasets, eg. the data-adaptive loss function proposed in Vu et al. 2022 "A Data-Adaptive Loss Function for Incomplete Data and Incremental Learning in Semantic Image Segmentation". Can the authors discuss why the teacher models are required to train the light-weight model to handle all anatomies?"
>
> - We agree with the reviewer that the benefit of data-adaptive loss is one way to train networks with partially labeled datasets. This has been added to the Introduction section and we consider adding experiments to compare with such methods. In our work, we build upon previous work from Kolbinger et al. [1] and use  one network with a shared encoder and separate decoders for each sub-dataset
>
> "The authors claim that while the teacher models were trained for 100 epochs, the student models required only 60. I assume these numbers were found empirically? Please explain how so. If the performance did not improve over 100 and 60 epochs respectively, can the authors address why the student models are faster to train than the teacher models?"
>
> - Our experiments have shown that training with an additional training signal through the Kullback Leibler divergence causes the student networks to diverge more quickly. We have therefore reduced the number of epochs from 100 to 60.
>
>
> "In the results section the authors say that "A uniform increase in segmentation performance can be observed across all eleven anatomies". However these detailed results are not presented in the manuscript. Perhaps the authors could include a more detailed evaluation table in the appendix to show anatomy-based evaluations?"
>
> - We thank the reviewer to highlight this important aspect. We added a more detailed evaluation table in the appendix.
>
> "The performance of the proposed model is questionable, based on the results in Table 1. Looking at the values, a reader can say, that models that have small number of trainable parameters (MiniNetv2) perform poorly, however they have a short inference time. [..] It is difficult to see what the model adds to the list of solutions in the field. Models trained with the KD-approach doesn't stand out in any of the metrics individually, only when looking at the metrics collectively. What do the authors think of collecting the results instead only for a certain model architecture, showing how the KD-approach and the other two learning methods impact performance and inference time? What is the benefit of a collective overall table such as Table 1 for all model performances?"
>
> - With our Multi-Teacher Knowledge Distillation approach we aim to increase segmentation performance across various network architectures. We added a more comprehensive evaluation by adding results from MiniNetv2 and ResNet18 to Figure 1 and Table 1.
>
>
>
>
> [1] Fiona R Kolbinger, Franziska M Rinner, Alexander C Jenke, Matthias Carstens, Stefanie
> Krell, Stefan Leger, Marius Distler, J ̈urgen Weitz, Stefanie Speidel, and Sebastian Boden-
> stedt. Anatomy segmentation in laparoscopic surgery: comparison of machine learning
> and human expertise–an experimental study. International Journal of Surgery, 109(10):
> 2962–2974, 2023.

---

### Official Review · Reviewer_eLLg · 2024-03-04

**Confidence:** 2
**Preliminary Rating:** 2

**Summary:**

This paper introduces a multi-teacher knowledge distillation approach to improve the performance of small and capable neural network which targets for real-time automatic anatomy segmentation task in laparoscopic surgery. This approach inlcudes training anatomy specific binary segmentataion models and using these models as teachers to guide the training of multi-anatomy student segmentation model. The authors perform experiment on the largest public dataset of laparoscopic image and show that their approch can improve the segmentation performance.

**Strengths:**

* This paper addresses an important question.
* The application scenatrio is clearly described and motivation of this work is introduced clearly.
* The method and schematic figure are clear and easy to understand.

**Weaknesses:**

* The novelty and the contribution of the paper seem limited. The models this paper used are directly from related papers. The Multi-Teacher knowledge distillation approach in general is also not novel. And the objective functions for training are very basic. From the experiment results, the proposed approach only improved the disc score compared with previous models. The model size, and inference time are the same, which make what the authors claimed in the title as “efficient” questionable.

* The paper content is not structured well. The authors use quite a lot of text to introduce the DEAD dataset, e.g section 2.1 and the 3rd paragraph in Introduction section. It is quite redundant since this dataset is not their work. The introduction section mainly focus on the motivation of the proposed approach and missing related works on multi-teacher knowledge distillation which is very relevant.

* The authors highlight the improvement of models based on segmentation performance.  From the results in Table 1, MiniNetv2 is small the inference time is the best, and has a potential for a big performance improvement with the proposed method. Why the author did not try their approach on MiniNetv2 to check how much the performance can improve with their proposed method?

**Detailed Comments:**

* Please clearly state the novelty of this work from the previous work. What has been done before and what are not.
* Please add more related works in teacher-student learning in segmentation such as "Improving Fast Segmentation With Teacher-student Learning" and differentiate your work from the others.
* Please consider adding the experiment results of MiniNetv2 or other models with the proposed approach to show its generality.

**Justification Of The Preliminary Rating:**

* First of all, the novelty and contribution of this paper seem limited.
* Some related works on multi-teacher knowledge distillation and its state-of-the-art application in segmentation are missing.
* The experiment results from current settings are not sufficient to support the generality of the proposed method.

**Questions To Address In The Rebuttal:**

* What are the results on DeepLabv3 with other backbones such as ResNet18 and MiniNetv2 ? Do they all have significant performance improvement with the proposed knowledge distillation approach?

---

> ### Author Response · Authors · 2024-03-18
>
> "The paper content is not structured well. The authors use quite a lot of text to introduce the DEAD dataset, e.g section 2.1 and the 3rd paragraph in Introduction section. It is quite redundant since this dataset is not their work."
>
> - The reviewer is right that the DSAD information are partially redundant across section 2.1 and 3rd paragraph in Introduction Section. We updated the Introduction Section and merely moved important and supplementary information from the section 2 (Method) to the section 3 („ Experimantal Setup“)
>
> "Please clearly state the novelty of this work from the previous work. What has been done before and what are not. / Please add more related works in teacher-student learning in segmentation such as "Improving Fast Segmentation With Teacher-student Learning" and differentiate your work from the others."
>
> - We agree with the reviewer, that the novelty of our work has not been stated clearly enough and especially a comparison with previous work could be explained in more detail.
> This has been added to the introduction section of our work.
>
> "What are the results on DeepLabv3 with other backbones such as ResNet18 and MiniNetv2 ? Do they all have significant performance improvement with the proposed knowledge distillation approach?"
>
> - We thank the reviewer to highlight this important aspect. We added further results to Table 1 and can show highest performance improvements for the MiniNetv2 when using our MT-KD approach. This strengthens the hypothesis that our proposed approach shows high generality.

---

### Author Response · Authors · 2024-03-18

Dear reviewers,
first of all, we want to thank you for the detailed and constructive reviews on our work. We are happy to see that most reviewers appreciate the important application scenario, our aim to make the code repository publicly available and the results of our learning approach.
However, the proposed directions to improve our work are especially helpful. This includes, among others, a more comprehensive evaluation of our approach to show its generality, as well as a comparison with existing work in the field of knowledge distiallation and especially a focus on the novelty of our method. In the following comments, we aim to answer your questions. All changes in the revised manuscript are highlighted in orange. Overall, we are very grateful for the detailed reviews of our work and hope that this will improve our future research.

---

> ### Author Response · Authors · 2024-03-22
> **Update 22.03.**
>
> Update 22.03.
>
> We updated the abstract in the manuscript to highlight the generality of our proposed approach and its applicability to partially labelled datasets.
> Additionally, we added Figure 3 to show qualitative segmentation results for different anatomies. We adapted the results section and discussed the qualitative results.
> All changes in the revised manuscript are highlighted in orange. We thank the reviewers for their assessment of the revised manuscript.

---

### Meta-Review · Area_Chair_ahKp · 2024-04-04

**Recommendation:** Accept (Poster)
**Confidence:** 4

**Metareview:**

Based on the improvements made by the authors in response to the reviewers' comments, the manuscript shows promising progress in addressing the concerns raised. The authors have taken steps to improve the clarity, readability and novelty of their work, and have conducted additional experiments to further validate their approach. Some concerns remain, particularly with regard to further validation of the proposed approach, but I still suggest accepting the paper for a poster presentation in MIDL.

---

### Decision · Program_Chairs · 2024-04-06

Accept (Poster)